# Selenium-Enriched Soy Protein Has Antioxidant Potential via Modulation of the NRF2-HO1 Signaling Pathway

**DOI:** 10.3390/foods10112542

**Published:** 2021-10-22

**Authors:** Xiaoli Zhao, Jinyan Gao, Astrid Hogenkamp, Leon M. J. Knippels, Johan Garssen, Jing Bai, Anshu Yang, Yong Wu, Hongbing Chen

**Affiliations:** 1State Key Laboratory of Food Science and Technology, Nanchang University, Nanchang 330047, China; x.zhao1@uu.nl (X.Z.); Gaojinyan@edu.ecu.cn (J.G.); baijin@ecu.edu.cn (J.B.); yanganshu@ncu.edu.cn (A.Y.); wuyong@ncu.edu.cn (Y.W.); 2Division of Pharmacology, Utrecht Institute for Pharmaceutical Sciences (UIPS), Utrecht University, 3584 CG Utrecht, The Netherlands; a.hogenkamp@uu.nl (A.H.); l.m.j.knippels@uu.nl (L.M.J.K.); johan.garssen@danone.com (J.G.); 3School of Food Science Technology, Nanchang University, Nanchang 330047, China; 4Sino-German Joint Research Institute, Nanchang University, Nanchang 330047, China; 5Global Centre of Excellence Immunology, Danone/Nutricia Research, 3584 CT Utrecht, The Netherlands

**Keywords:** Se, Se-enriched soy protein, antioxidant, NRF2, antioxidant enzyme

## Abstract

Selenium (Se)-enriched proteins are an important dietary source of Se for humans; however, only a few Se-enriched proteins have been identified. In the present study, we tested for potential antioxidant activity by Se-enriched soy protein, both in vitro and in vivo. Se-enriched soy protein isolate (S-SPI) was shown to have a higher free radical scavenging ability compared to ordinary soy protein isolate (O-SPI). Furthermore, Caco-2 cell viability was improved by S-SPI at low doses, whereas O-SPI did not. In addition, S-SPI was shown to inhibit oxidative stress via modulation of the NRF2-HO1 signaling pathway, upregulating the expression of downstream antioxidant enzymes (GPx, SOD). To further study the antioxidant capacity of S-SPI, BALB/c female mice were given oral gavages with 0.8 mL of S-SPI or O-SPI (5 g/kg/d, 20 g/kg/d and 40 g/kg/d) or saline as control. Hepatic GPx and SOD activity increased with increasing S-SPI dosage, but not with O-SPI. Taken together, our results suggest that Se-enriched soy protein has a high antioxidant ability and may be used as a dietary supplement for people with oxidative dam-age-mediated diseases.

## 1. Introduction

Selenium (Se) is an essential micronutrient as selenoproteins participate in physiological processes including antioxidation, thyroid hormone metabolism, cancer and inflammatory responses [1]. Se is one of the most important micronutrients as it exerts multiple and complex effects on human health. Se was known as an environmental toxicant for livestock with potential harmful effects for human health [2]. Today it is apparent that Se can have toxic effects, usually at high doses [3], but also behaves as an essential dietary micronutrient at low doses [4]. Dietary requirements for Se in humans have been estimated to range from 47 to 105 μg Se per day [5]. In experimental animal studies, chronic Se deficiency (<0.2 mg/kg diet for 6 weeks) has been shown to result in less robust immune responses to viruses and tumors compared to Se-adequate controls [6,7,8]. In humans, inadequate Se status (Serum Se values < 100 ng/mL) is associated with increased risk of cardiovascular disease [9], cancer [10] and impaired immune function [11].

Our previous study indicated that Se species of SPIs are naturally present in plants, with a recording of the Se content of S-SPI and O-SPI are 339.25 μg/kg and 37.12 μg/kg, respectively [12]. The amounts of Se present in the isolates used in the study. It is generally believed that organic forms of Se have lower toxicity than inorganic forms. Furthermore, organic Se is more easily absorbed in the intestine and transported via the bloodstream into organs to be used for physiological functions. Organic forms of Se can be found in fruit, vegetables and yeast because the selenate and selenite forms of Se are readily absorbed by plants and metabolically converted in the chloroplast to organic Se compounds, such as SeMet and SeCys [13]. SeMet is the main chemical form of dietary Se [14]. Ingested SeMet from foods or nutritional supplements is absorbed via the intestine and undergoes trans-sulphuration to produce SeCys [15]. Incorporation of SeCys, through recoding of the UGA stop codon, creates a unique class of proteins such as gluthathione peroxidase (GPx) and selenoprotein P (SELENOP), which are secreted into plasma and serve as a source of Se to other tissues of the body [16]. This mammalian enzyme and selenoprotein, is widely distributed in mammalian tissues including the liver [17]. GPx is known to be important in defense against organic peroxide and hydrogen peroxide (H_2_O_2_)-induced damage to cell membranes [18]. 

The use of molecular oxygen by aerobic organisms inevitably leads to the formation of many oxygen-containing reactive substances, which are collectively called reactive oxygen species (ROS) [19]. ROS quickly accumulates in cells, resulting in thermal stress which leads to oxidative damage of the tissue and, ultimately, accelerated death of the organism [20]. These significant damaging effects to tissue caused by ROSs highlights the importance of cellular processes that prevent the accumulation of O_2_^−^. Antioxidant enzymes including GPx family, superoxide dismutase (SOD) and catalase (CAT) play an important role in defending tissue and organs against over-production of ROS [21]. SOD is considered the first line of defense and the main regulator of ROS levels by catalyzing the conversion of superoxide radicals to H_2_O_2_ and molecular oxygen. GPx and CAT then convert H_2_O_2_ into oxygen and H_2_O [22]. The antioxidant activity of Se-containing proteins has now been shown in various animal models. For example, Rui Zeng (2019) found that Se-containing protein treatment had a positive influence on liver hepatocytes in mice, which are involved in defense against ROSs and support the function of the enzymatic antioxidant system [23]. Similarly, Li Guang (2011) indicated that supplementation of Se-yeast can improve serum oxidant status by increase antioxidant enzymes activity in growing male goats [24]. However, the mechanism by which Se-containing protein enhances antioxidant activity is still unclear.

We have previously shown Se enrichment can strengthen protein against oxidant damage, playing an important role in maintaining the conformation (aggregation and partial unfolding) of Se-enriched soy protein under AAPH-induced oxidative stress [12]. However, the antioxidant effects of Se-enriched soy protein are rarely reported, and their antioxidative mechanisms have yet to be clearly identified. In the present study, we compared the free radical scavenging ability and effects on antioxidant-related enzyme activity and the NRF2 signaling pathway of a Se-enriched soy protein isolate (S-SPI) with ordinary soy protein isolate (O-SPI). 

## 2. Materials and Method

### 2.1. Materials

Se-enriched soybeans were obtained from Natural Se Base of Fengcheng, Jiangxi, China. Bicinchoninic acid [25] kits were purchased from Beyotime Institute of Biotechnology (Shanghai, China). Assay kits for the determination of malondialdehyde [26] concentrations and SOD and GPx activities were purchased from Beyotime Institute of Biotechnology (Shanghai, China). Antibodies for NRF2 (D1Z9C) XP^®^ rabbit mAb (Alexa Fluor^®^ 647), HO-1 monoclonal antibody (HO-1) (FITC) and monoclonal antibody NQO-1 monoclonal antibody (A180) [PE] were purchased from Cell Signaling Technology (Danvers, MA, USA). Ultrapure water used in all experiments was supplied by a Milli-Q water purification system from Millipore.

### 2.2. Preparation of Soybean Protein Isolate 

Soybean protein isolate (SPI) was prepared from Se-enriched and ordinary soybeans as previously described [12]. Briefly, defatted soybean flour was dispersed in distilled water (1:10, *w/v*), adjusted to pH 8.0 with 1 M NaOH, stirred for 3 h at room temperature and then centrifuged at 5530× *g* for 20 min. The supernatant was precipitated at pH 4.5 using 1 M HCl at 4 °C for 1 h and separated by centrifugation at 5530× *g* for 20 min. The sediment was then dissolved in deionized water and adjusted to pH 7.5 before dialyzing with deionized water for 24 h at 4 °C. The resulting protein isolates from ordinary soybean and Se-enriched soybean were freeze-dried and stored at 4 °C.

The Se content of S-SPI is 339.25 μg/kg and O-SPI is 37.12 μg/kg.

### 2.3. Superoxide Anion Scavenging Activity

The superoxide anion scavenging activity of SPI was measured using a pyrogallol autoxidation system with some modifications [27]. A reaction solution containing 80 μL of pyrogallol (10 mM) and 4.5 mL of Tris/HCl/EDTA buffer (50 mM, pH 8.0) was prepared, to which 0.5 mL of O-SPI or S-SPI (1–5 mg/mL) were added. The solution was incubated at room temperature for 30 min, after which 200 µL of the solution was transferred to a 96-well microplate to measure the absorbance at 365 nm with a microplate reader (Bio-Rad, Hercules, CA, USA). Scavenging activity was calculated according to the following equation:Scavenging effect (%) = [1 − (*A*1 − *A*2)/*A*0] × 100
where *A*0 is the absorbance of the control (without sample), *A*1 is the absorbance in the presence of the sample, and *A*2 is the absorbance of the sample without pyrogallol.

### 2.4. Measurement of Reducing Power

To assess/compare the antioxidant capacity of O-SPI and S-SPI, the reducing power of S-SPI and O-SPI was determined according to the method of Li [28] with some modifications. In brief, 1 mL of protein (1–5 mg/mL) was mixed with 3 mL of 0.5 M sodium phosphate buffer (pH 6.6) and 2.5 mL of 1% potassium ferricyanide [K_3_Fe[CN]_6_]. The reaction mixtures were incubated in a water bath at 50 °C for 20 min, after which the mixtures were quickly cooled to room temperature and added 2.5 mL of 10% TCA. The mixtures were centrifuged at 3000× *g* for 10 min at 25 °C. The supernatant (2.5 mL) was then mixed with 0.5 mL of 0.1% FeCl_3_ and 2.0 mL of distilled water, and absorbance was measured at 700 nm.

### 2.5. Cell Culture and Treatment

Cell culture was conducted according to García-Nebot [29] with some modifications which is based on the conversion of MTT to formazan crystals by mitochondrial dehydrogenases. Caco-2 human intestinal cells were grown in DMEM medium (Gibco, Burlington, ON, Canada) with 10% fetal bovine serum (FBS; Hyclone Co., Logan, UT, USA), incubated at 37 °C in 5% CO_2_; the medium was replaced every 2 days. Cells were seeded onto 96-well plates at 3 × 10^4^ cells/plate and were allowed to equilibrate for 24 h prior to experiments. H_2_O_2_ was dissolved in serum-free DMEM at 400 mM. For the oxidative stress experiments, cells were incubated with different concentrations of O-SPI or S-SPI (0.1, 0.2, 0.3, 0.4, 0.5 mg/mL) for 12 h; the induction of oxidative stress was carried out by exposure to a 400 mM H_2_O_2_ solution in the culture medium for 6 h (37 °C /5% CO_2_/95% relative humidity).

### 2.6. Determination of Cell Viability

Cell viability was determined with the thiazolyl blue tetrazolium bromide (MTT) assay, which is based on the conversion of MTT to formazan crystals by mitochondrial dehydrogenases. After incubation with SPI and induction of oxidative stress in 96-well plates, media was discarded and replaced with serum-free media containing 0.5 mg/mL of MTT. Plates were further incubated for 3 h at 37 °C in 5% CO_2_. The MTT solution was subsequently removed and cells were lysed with 150 mL DMSO. Plates were then placed on an orbital shaker for 10 min before measurement of absorbance at 570 nm using a microplate reader. Absorbance values were expressed as a percentage of untreated control cells (control = 100%).

### 2.7. Flow Cytometry Analysis of the NRF2-HO1 Signaling Pathway in Caco-2

After incubation with SPI and H_2_O_2_, Caco-2 cells were collected, and cell-concentrations were adjusted to 4 × 10^6^ cells/mL. One-hundred microliters of the cell suspension were transferred into a PE tube and 1 mL flow cytometry staining buffer was added. After fixation and permeabilization (eBioscience), intracellular proteins were labeled with the corresponding mAbs conjugated with fluorescent molecules, according to the manufacturer’s instructions. Flow cytometry was performed on a BD FACSCalibur and analyzed with FlowJo V10 software.

### 2.8. Animals and Experimental Design

6–8-week-old female BALB/c mice (18.0 ± 2.0 g body weight) were provided by Hunan Slac Jingda Laboratory Animal Co. (Changsha, China, Certificate number: SCXK (Xiang) 2014−0011). All mice used in this study were cared for in accordance with the Guidelines for the Care and Use of Laboratory Animals published by the U.S. National Institute of Health (NIH Publication 85–23, 1996), and all experimental procedures were approved by the Animal Care Review Committee, Nanchang University (permission number: SYXK (Gan) 2013–0004). Mice were housed in an animal room at 23 ± 1 °C and 50 ± 5% relative humidity, under a 12/12 h of light-dark cycle and provided with ad libitum access to water and commercially available Se-free rodent chow. After a 7 d acclimatization period, all mice were randomly divided into seven groups (*n* = 6 per group) including one normal control (NC) group and six SPI dosage groups. The gavage dosage based on the dietary requirements for Se in adult humans (47 to 105 μg Se /day) and lowest requirement (15 μg Se/day). Then, 15, 47 and 105 μg Se/day for 70 kg adult converted to 20 g mice dosage in S-SPI are around 5, 20 and 40 g protein/kg/d. Based on this calculation, we calculated dosage in our experiment are low dosage (5 g/kg/d L-O-SPI or L-S-SPI), medium dosage (20 g/kg/d M-O-SPI or M-S-SPI) and high dosage (40 g/kg/d H-O-SPI or H-S-SPI); saline (NC group) or SPI were administered by gavage every day for 28 d. At the end of the experiment (day 28) the animals were killed; spleen, stomach, liver and kidney were dissected to measure the Se content in main organs. A section of the liver was used for the determination of oxidation-related enzymes. 

### 2.9. Measurement of Se Concentration in Tissues

Se concentration in mouse tissues was measured according to Chen [30] with some modifications. Samples (100 mg) were digested with 10 mL of HNO_3_ and HClO_4_ mixture (9:1, *v/v*) at 150 °C for 2.5 h. After cooling, 5 mL of 6 M HCl were added to the digested samples. The digested solution was allowed to cool again, diluted with ultrapure water to a final volume of 25 mL, and analyzed by atomic fluorescence spectroscopy (Model AI 3300, Aurora Technologies, North Vancouver, BC, Canada). 

### 2.10. Measurement of Superoxide Dismutase (SOD) and Glutathione Peroxidase (GPx) Activities

For in vitro experiments, Caco-2 cells were collected after incubation with SPI and H_2_O_2_ and lysed with Triton-100. Homogenates were centrifuged (600× *g*, 4 °C, 10 min) and supernatants were collected. For in vivo experiments, livers were dissected and washed in ice-cold PBS. Ten percent (w/v) liver homogenates were prepared by homogenization in ice-cold physiological PBS and centrifuged (1600× *g*, 4 °C, 10 min); homogenates were collected. SOD and GPx activity were analyzed with commercially available kits according to the manufacturer’s instructions.

### 2.11. Statistical Analysis

All in vitro determinations were performed in triplicate and results were presented as mean ± SEM. Statistical analysis was performed using GraphPad Prism 8 software (GraphPad Company, San Diego, CA, USA). Statistical comparisons were made by one-way analysis of variance (ANOVA) using GraphPad Prism 8. Statistical significance was set at *p* < 0.05.

## 3. Results

### 3.1. S-SPI Effectively Increases Superoxide Anion Scavenging Ability and Reducing Power of Fe^3+^

Low doses of S-SPI (1 to 5.0 mg/mL) increased the scavenging rate of O_2_^−^ radicals from 15% to 32%, whereas no such dose-dependent increase in O_2_^−^ scavenging was observed for O-SPI (14–17%) (Figure 1A). Next, we assessed the reducing power of SPI based on the competence of antioxidants in converting potassium ferricyanide (Fe^3+^) to potassium ferrocyanide (Fe^2+^). Reducing power was higher in S-SPI compared to O-SPI in same protein concentration (Figure 1B).

### 3.2. S-SPI Treatment Protects Caco-2 Cells from Oxidative Stress Induced by H_2_O_2_

#### 3.2.1. S-SPI Treatment Improves Cell Viability in H_2_O_2_-Exposed Caco-2 Cells

To investigate whether SPI could reduce H_2_O_2_-induced oxidative stress and have cytoprotective effects, Caco-2 cells were pretreated with varying concentrations of O-SPI or S-SPI (0.1 to 0.5 mg/mL) then exposure to 400 mM H_2_O_2_ for 4 h. Exposure to H_2_O_2_ resulted in a significant reduction (up to 60%) in cell viability compared to the controls (Figure 2). The most effective S-SPI treatment dose was 0.4 mg/mL, with cell viability declining when SPI concentrations further increased to 0.5 mg/mL. An upward trend was observed for O-SPI concentration and cell viability, but the increase in cell viability was not statistically significant (Figure 2). These results indicate the potential for S-SPI to improve cell viability under oxidative stress compared to the same dose of O-SPI.

#### 3.2.2. S-SPI Treatment Increases Activity of Key Antioxidant Enzymes SOD and GPx in Caco-2 Cells

To analyze effect of O-SPI and S-SPI on the activity of the enzymes in vitro, the activity of two enzymes (SOD and GPx) were investigated. Exposure to H_2_O_2_ resulted in down-regulation of SOD and GPx activity of Caco-2 cells, but an increase in activity was observed in cells pre-treated with O -SPI and S-SPI (Figure 3). Cells treated with S-SPI (0.2–0.3 mg/mL) expressed higher activity levels of SOD compared to untreated cells, but cells treated with O-SPI did not express a significantly different level of SOD activity compared to untreated cells (Figure 3A). S-SPI treatment, but not O-SPI-treatment, also resulted in increased GPx activity although this effect was only observed at 0.3–0.5 mg/mL (Figure 3B). These results show that Se-enriched protein can effectively enhance endogenous antioxidant enzyme activity.

#### 3.2.3. Suppression of Oxidative Stress in H_2_O_2_-Exposed Caco-2 Cells by S-SPI Is Related to Increased Protein Levels of the NRF2 Pathway

To further study the mechanism underlying the protective effect of S-SPI in H_2_O_2_-exposed Caco-2 cells, we examined the effect of S-SPI on the NRF2 signaling pathways by quantifying the level of expression of the proteins HO-1, NQO1 and NRF2 (Figure 4). Overall, treatment with 0.1–0.3 mg/mL concentration of S-SPI increased expression of HO-1, NQO1 and NRF2 levels compared to untreated H_2_O_2_-exposed cells (control). However, at higher concentrations both S-SPI and O-SPI lowered expression levels of HO-1, NQO1 and NRF2 (Figure 4). These results indicate that there is a dose-dependent effect of SPI on the expression of proteins in the NRF2 signaling pathway.

### 3.3. S-SPI Supplementation Effectively Enhances Antioxidant Activity in BALB/c Mice

#### 3.3.1. Supplementation with S-SPI Significantly Increases Tissue Levels of BALB/c Mice

To evaluate Se-levels in murine tissues after four weeks of daily intragastric administration of O-SPI and S-SPI, we analyzed Se concentration in liver, kidney, spleen and stomach of the animals (Table 1). Overall, an increase in Se levels was observed in the analyzed tissues of all mice fed an SPI-supplemented diet compared to control mice (Table 1). However, for O-SPI this difference was only statistically significant for renal tissues of animals receiving the medium and high doses, whereas significant differences in Se levels were observed in all tissues when M-S-SPI and H-S-SPI were compared with tissue from control animals. In S-SPI-supplemented animals, the increase in Se levels was most apparent in the liver, with a 4.3-fold increase in the H-S-SPI group (1.34 mg/kg Se) compared to control animals (0.31 mg/kg Se). Similarly, Se levels were significantly and dose-dependently increased in kidneys of S-SPI treated mice. Compared to the controls, a twofold increase in renal Se levels was observed in mice treated with H-S-SPI (0.46 mg/kg and 0.92 mg/kg, respectively). Renal Se levels were also increased in the M-O-SPI and the H-O-SPI groups (0.69 mg/kg and 0.66 mg/kg, respectively) compared to controls (0.46 mg/kg). These results indicate that treatment with Se enriched protein can increase free Se levels in murine tissues, particularly in the liver.

#### 3.3.2. SPI Supplementation Increases Activity of Hepatic SOD and GPx of BALB/c Mice

As the biggest increase in Se levels were observed in the liver, we analyzed the activity levels of SOD and GPx activity in this tissue in vivo (Figure 5). SOD activity was observed to increase in liver tissue of mice in all SPI-treatment groups, compared with the control group (Figure 5A), and the increase appeared to be dose-dependent, although SOD activity declined in animals treated with H-O-SPI. This outcome is in line with the observed levels of Se in the livers of O-SPI treated animals (Table 1). However, when similar doses were compared, SOD activity was higher in liver tissue of low and high dose S-SPI treated animals compared to O-SPI treated animals. In addition, we analyzed the activity of GPx (Figure 5B) and found an increase in medium and high dose S-SPI treated mice compared to control and O-SPI treated animals. 

## 4. Discussion

In the present study, we investigated the free radical scavenging activity of Se-enriched soy protein by measuring superoxide anion scavenging activity and its reducing power of Fe^3+^. Additionally, effects of S-SPI in the regulation of key NRF2-regulated antioxidant enzymes in human intestinal Caco-2 cells were measured. Both S-SPI and O-SPI were observed to have superoxide radical scavenging ability (Figure 1). Yufang (2015) indicated that SPI and SPI particles are observed to reduce hydrogen peroxide [31], likely due to the protein’s capacity to inhibit lipid oxidation through multiple pathways including inactivation of ROS, scavenging free radicals and reduction of hydroperoxides [32]. Furthermore, the radical scavenging abilities demonstrated by SPIs have been attributed to the presence of small peptides and C-terminal aromatic tyrosine residues in their molecular structure [33]. However, in the present study, Se-enrichment of SPI species contributed to their higher antioxidant activity compared to controls and O-SPIs (Figure 1). This is in line with the observation that proteins extracted from Se-enriched mushrooms exhibit increased scavenging activity against superoxide anions and hydroxyl radicals [34], and similar results were obtained for Se-enriched peanut leaf protein samples [35]. 

Previous studies have postulated various mechanisms by which S-SPIs have enhanced antioxidant activity. Firstly, Se compounds were found to exert free radical scavenging activity through the central free radical enzymes, which increased the antioxidant activity of the protein [36]. Secondly, the major Se compounds in the soy protein: selenomethionine (SeMet) and selenocysteine (Secys), were found to be central to increased antioxidant activity [37]. Furthermore, although the majority of Se in soy bean is organic, Se(VI) and Se (IV) were also detected in extracts and found to enhance hydroxyl radical scavenging activity and reducing power [37]. Finally, it was found that Se amino acids react with the hydrated free radicals (•OH, •H) to form stable compounds before they destroy the biological macromolecules, and they easily form stable positive ion free radicals which possess strong antioxidant activity [38]. Thus, antioxidant mechanism of Se-enriched soy protein may relate one or multiple of those mechanisms. 

Caco-2 cells are important intestinal cell types and are used for evaluations of antioxidant capacity of food and natural extracts [39]. We exposed untreated and pre-treated (with S-SPI and O-SPI) Caco-2 cells to H_2_O_2_-type oxidants to investigate the antioxidant capacity and possible mechanisms of S-SPI (Figure 2 and Figure 3). We found that S-SPI could improve the activity of SOD and GPx via activating NRF2 expression, which could explain the underlying antioxidative mechanism (Figure 3). SOD is one of the antioxidative enzymes that play a crucial role in scavenging ROS and maintaining homeostasis of oxidation-antioxidation in the body [40]. GPx is an antioxidant enzyme that can defend against severe free radical oxidation [41]. Therefore, the reason for and the degree of cell injury could be reflected by the activity of SOD and GPx. Consistently, decreases of GPx and SOD activity in cells were observed after H_2_O_2_ treatments. Pretreatment with S-SPI increased cell viability and improved SOD and GPx activity compared to the control and O-SPI groups. Although the difference in enzymatic activity was significant, this may also be explained by the fact that the cell viability was significantly increased as a result of S-SPI treatment. Unfortunately, the assessment of the cell viability does not allow for determination of cell numbers, which limits the interpretation of the difference in enzymatic activity.

NRF2 is an important regulator of the antioxidant of the cell defense system as the NRF2 pathway controls detoxifying enzymes, antiapoptotic proteins, and proteasomes. Antioxidant genes regulated by NRF2 such as NADPH dehydrogenase quinone 1( NQO1) and HO-1, are known to be upregulated by the pathway [42] and further improve antioxidant activity. In the present study, we found up-regulation of both NRF2 and HO-1 when cells were pretreated with low doses of SPI (Figure 4), and pretreatment with S-SPI also significantly upregulated protein expression of NQO1 with highest expression of all three proteins in the 0.3 mg/mL dose group. Protein expression of NRF2, NQO1 and HO-1 decreased in cells treated with 0.4–0.5 mg/mL of both O-SPI and S-SPI, but at this dose, concentration of downstream antioxidant enzymes increased. These dose-dependent effects of S-SPI are not yet clear.

As well as studying the in vitro effects of SPI, we also investigated the effects of supplementing BALB/c mice with SPI. We observed that Se levels were highly elevated in the liver, in accordance with a study that showed a clear dose response distribution of Se in tissues and the highest Se concentration was recorded in the liver [43]. Mamoru et al. (2013) also found that the Se concentration was higher in the liver of Se-deficient Tg2576 mice when the mice were fed with a Se-supplemented diet [44]. Previously, it was reported that Se depletion was most prominent in the lungs for Txnrd1 or Txnrd2 hemizygous mice under Se-adequate conditions, and most prominent in the kidneys under selenium-deficient conditions [45]. Similarly, Oster et al. (1987) found the highest Se concentrations in the kidneys of German Adults [46], suggesting that the distribution of Se is dependent on the subjects and their conditions. We also assessed the enzymatic activity of SOD and GPx in liver cells of the mice. We found that S-SPI treated mice enhanced the activities of the antioxidant enzymes SOD and GPx (Figure 5). The activity of GPx was significantly improved. Taken together, our in vitro and in vivo data suggest that S-SPI may prevent oxidative damage by scavenging free radicals and aiding the NRF2 signaling pathway by upregulating downstream antioxidant enzymes.

## 5. Conclusions

These results suggest that Se-enriched soy protein have stronger free radical scavenging activity compared to ordinary soy protein. Meanwhile, compared with ordinary soy protein, Se-enriched soy protein effectively improved cell viability and increased antioxidant enzymes activity under the oxidant stress condition. Thus, antioxidant activity of soy protein can be improved by Se enrichment. These results suggest that the antioxidant activity of soy protein can be improved by Se enrichment. Se-enriched soy protein reduced oxidation by activating the NRF2 pathway, leading to increased levels of SOD and GPX levels, which are crucial influencing factors for reduction of oxidative stress. Thus S-SPI could be used as a dietary supplement of protein for potential health benefits.

## Figures and Tables

**Figure 1 foods-10-02542-f001:**
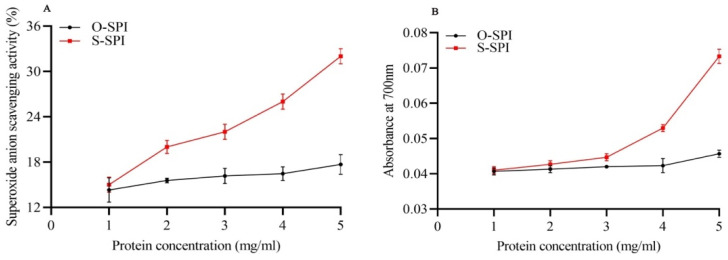
Superoxide anion radical scavenging ability (**A**) and reducing power (**B**) of S-SPI. Values are expressed as mean ± SEM (*n* = 3), *p* ≤ 0.05.

**Figure 2 foods-10-02542-f002:**
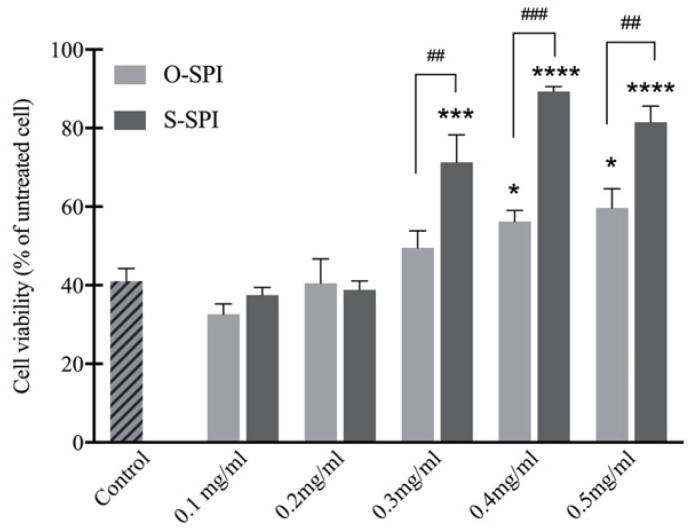
In vitro effect of S-SPI on cell viability as measured by MTT conversion (%). Caco-2 cells were incubated with 0.1–0.5 mg SPI/mL for 12 h and then treated with 400 μM H_2_O_2_ for 6 h. Values expressed as mean ± SEM (*n* = 3). Significant differences between control group and other group are indicated by * *p* < 0.05, **** *p* < 0.001, **** *p* < 0.0001. Differences between O-SPI and C-SPI under same concentration are indicated by ^##^ *p* < 0.01, ^###^ *p* < 0.001. Differences are analyzed with a two-way analysis of variance (ANOVA) followed by a Tukey test.

**Figure 3 foods-10-02542-f003:**
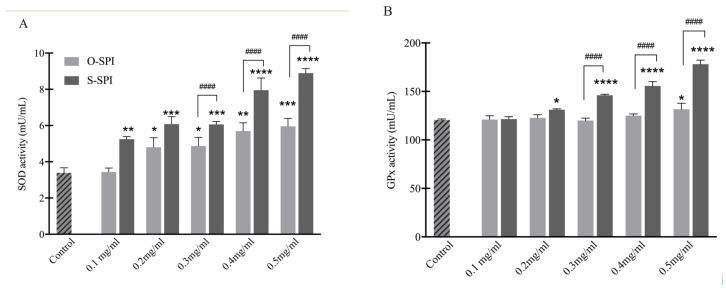
In vitro effects of S-SPI on antioxidant enzymatic activity of SOD (**A**) and GPx (**B**). Caco-2 cells were pretreated with different concentrations of SPI (0.1–0.5 mg/mL) for 12 h min and then treated with 400 μM H_2_O_2_ for 6 h. Values are expressed as mean ± SEM (n = 3). Significant differences between control group and other group are indicated by * *p* < 0.05, ** *p* < 0.01, *** *p* < 0.001, **** *p* < 0.0001. Differences between O-SPI and C-SPI under same concentration are indicated by ^####^ *p* < 0.0001. Differences are analyzed with a two-way analysis of variance (ANOVA) followed by a Tukey test.

**Figure 4 foods-10-02542-f004:**
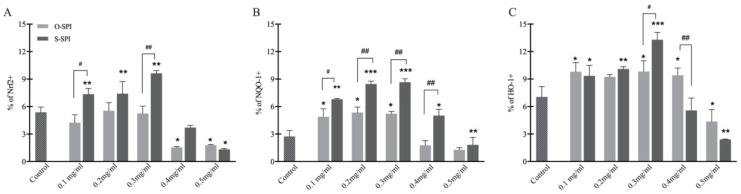
O_2_ -induced changes in the nuclear accumulation of Nrf2 and the expression of Nrf2-related in Caco-2 cells. Flow cytometry analysis of Nrf2 (**A**), NQO-1 (**B**), and HO-1 (**C**). Values are expressed as mean ± SEM (n = 3). Significant differences between control group and other group are indicated by * *p* < 0.05, ** *p* < 0.01, *** *p* < 0.001. Differences between O-SPI and C-SPI under same concentration are indicated by ^#^
*p* < 0.05, ^##^ *p* < 0.01. Differences are analyzed with a two-way analysis of variance (ANOVA) followed by a Tukey test.

**Figure 5 foods-10-02542-f005:**
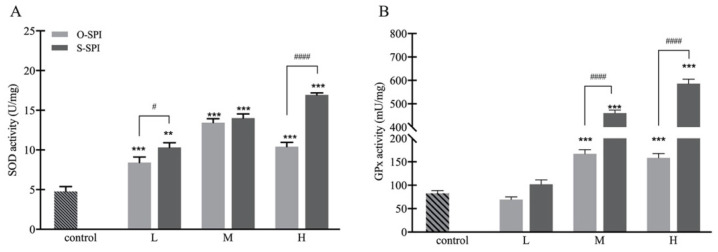
In vivo effects of S-SPI on hepatic antioxidant enzymatic activity of SOD (**A**) and GPx (**B**). BALB/c female mice were gavaged with low (L; 5 g/kg/d), medium (M; 20 g/kg/d), or high (H; 40 g/kg/d) dosages of SPI for 4 weeks. Values are expressed as mean ± SEM (n = 5). Significant differences between control group and other group are indicated by ** *p* < 0.01, *** *p* < 0.001. Differences between O-SPI and C-SPI under same dose are indicated by ^#^
*p* < 0.05, ^####^ *p* < 0.0001. Differences are analyzed with a two-way analysis of variance (ANOVA) followed by a Tukey test.

**Table 1 foods-10-02542-t001:** Se accumulation in different organ in mice after treated with different dose of protein. Different superscript letters indicate significant difference (*p* < 0.05) Se for each group.

	Control mg/kg	L-S-SPI mg/kg	M-S-SPI mg/kg	H-S-SPI mg/kg	L-O-SPI mg/kg	M-O-SPI mg/kg	H-O-SPI mg/kg
Liver	0.31 ± 0.09	0.61 ± 0.06	0.87 ± 0.05 ^b^	1.34 ± 0.15 ^a^	0.51 ± 0.03	0.62 ± 0.03	0.43 ± 0.05
Kidney	0.46 ± 0.03	0.54 ± 0.05	0.77 ± 0.03 ^b^	0.92 ± 0.05 ^a^	0.53 ± 0.03	0.69 ± 0.02 ^c^	0.66 ± 0.03 ^c^
Spleen	0.17 ± 0.04	0.35 ± 0.01 ^c^	0.40 ± 0.03 ^b^	0.51 ± 0.08 ^a^	0.29 ± 0.05	0.32 ± 0.04	0.31 ± 0.04
Stomach	0.07 ± 0.05	0.13 ± 0.01	0.15 ± 0.01 ^b^	0.18 ± 0.01 ^a^	0.12 ± 0.02	0.13 ± 0.02	0.12 ± 0.02

## Data Availability

The authors will send detailed data and calculations on request.

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
