# Peer review of "Selenium-Enriched Soy Protein Has Antioxidant Potential via Modulation of the NRF2-HO1 Signaling Pathway"

_foods, 2021, doi:10.3390/foods10112542_

Round 1
Reviewer 1 Report
The paper by Hongbing Chen and colleagues is focused on Se-enriched proteins from soy and on their antioxidant effect. The paper has formatting issues (see for example affiliations and spacing between keywords) and, in general, it must undergo major revision before resubmission. I report my comments below.
Abstract: too many abbreviations have been used. The abstract should be clear and concise to facilitate the reader. The first sentence must be rephrased since singular forms do not match (“Selenium (Se)-enriched protein is an important dietary source of selenium”).
Introduction: this section is quite long. Moreover, the authors should better specify the role of selenium as a micronutrient or as involved in a protein. Throughout the manuscript, the authors should check the use of apex and pedices (e.g. O2-). Sentences such as the following are not clear and must be rephrased: “We have previously shown that Se-enrichment effectively “protects” the antioxidant”.
Materials and methods: methods are adequately described and correctly referenced.
Results: I suggest the authors to improve and simplify Fig.5, as there are many symbols and may not be so clear for the reader.
Discussion: the first paragraph is rather generic, and I would suggest the authors to focus more in the molecular mechanism involved in the antioxidant effect. Is it connected with the Tyr content? Does it correlate with selenium? Moreover, I think the reader would appreciate an excursus on protein sequence and structure to better undertend the position and role of Se-aminoacids.
Conclusions: this part is very brief and results may be resumed in a more detailed fashion.
Some minor comments:
- Check the style of references
- Check punctuation, as in Table 1 (, used sometimes instead of .)
Reviewer 2 Report
Main remarks:
1) The main result of an effect of selenium-supplemented soya protein on CACO cell viability is convincing. It is a pity that you did not add two comparison groups: selenium as selenate or selenite versus no selena(i)te (both without soya protein). It is difficult through your article to differenciate the part of effect due to soya protein and the part due to selenium. Particularly, the curious effect of increasing doses soya protein (± selenium) on NF2 cannot be explained: is it due to selenium or to soya protein?
2) If cells are more viable with selenium, it is logical that superoxide dismutase and gluathione peroxidase are increased. Again, here also, a "control protein measurement" such as albumin or prealbumin in CACO cells not affected by selenium or soya protein should be useful to distinguish the variation of GPX and SOD due to variation of viability, or due to variation of selenium.
3) Some references to selenium molecular biology in mammals are lacking (look for example to review by Marla J. Berry or Josef Kohrle)
Please find also some suggestions of improvements in the attached Word document.
Ideally, if possible, a few weeks to conduct new experiments proposed in point 1. should greatly improve the impact of this work.

Round 2
Reviewer 1 Report
The authors answered to the comments previously raised on the manuscript. Thus, the paper meets the standard for publication in this Journal.
Reviewer 2 Report
Remarks have been added in the discussion on difficulty related to the enzyme GPX-SOD variation possibly associated with decreased viability of the CACO cells. According to the authors, it was not feasible to add some more experiments on inorganic selenite or selenate, to compare with organic Se-soya. For info (this is not a remark), but you know likely this paper : your manuscript in these COVID times revitalises also previous interesting experimental work on the protective effect of selenium on FLU 21 years ago (Melinda Beck & al., 2001, Selenium deficiency increases the pathology of an influenza virus infection, FASEB J) : maybe there is some overlap with your work.